# SHAPING MONOTONIC NEURAL NETWORKS WITH CONSTRAINED LEARNING

## ABSTRACT

The monotonicity of outputs of a neural network with respect to a subset of inputs is a desirable property that provides an important tool to explore the interpretability, fairness, and generalizability of the designed models, and underlies many applications in finance, physics, engineering, and many other domains. In this paper, we propose a novel, flexible, and adaptive learning framework to induce monotonicity of neural networks with general architectures. The monotonicity serves as a constraint during the model training, which motivates us to develop a primal-dual learning algorithm to train the model. In particular, our framework provides an interface to trade off between probability of monotonicity satisfaction and overall prediction performance by introducing a chance constraint, making it more flexible for different application scenarios. The proposed algorithm needs only small extra computations to continuously and adaptively enforce the monotonicity until the constraint is satisfied. Compared to the existing methods for building monotonicity, our framework does not impose any constraints on the neural network architectures and needs no pre-processing such as tuning of the regularization. The numerical experiments in various practical tasks show that our method can achieve competitive performance over state-of-the-art methods.

## 1 INTRODUCTION

Deep neural networks have significantly promoted the development of many real-world domains such as finance, physics, engineering (He et al., 2016; LeCun et al., 2015; Vaswani et al., 2017), and drastically reshaped various fields like computer vision (You et al., 2017; Minaee et al., 2021), natural language processing (Otter et al., 2020), autonomous driving (Kiran et al., 2021). Shaping the neural network models with prior knowledge, the monotonicity dependence for example, is a desideratum that can help improve the interpretability and generalizability of the models (Feelders, 2000; Rieger et al., 2020; Liu et al., 2020). The incorporation of monotonicity with respect to some parts of inputs can also help the trained models produce legal, reliable, and safe results. For instance, in the admission decisions on applicants using an machine learning model, the machine learning model is expected to select the student with higher scores when all other factors are equal (Liu et al., 2020). Similar monotonicity requirements are prevalent in engineering. For example, when designing controllers for safe-critical systems, e.g., autonomous vehicles (Chen et al., 2024; Bojarski et al., 2016) and power systems (Cui et al., 2023a;b), the stability of the physical system often relies on monotonicity of the controllers. Enforcing monotonicity in the controller models is important to ensure the safety and reliability of such physical systems.

Incorporating the monotonicity constraints has been widely studied in traditional machine learning over the past decades (Archer & Wang, 1993; Potharst & Feelders, 2002; Doumpos & Zopounidis, 2009; Chen et al., 2015; Bartley et al., 2019), and recent research has extended the attention to inducing the monotonicity constraints into complex neural network models (Lang, 2005; You et al., 2017; Milani Fard et al., 2016; Liu et al., 2020). Existing methodologies can generally be categorized into two groups:

*1) Monotonicity by constructed architectures.* This group enforces the monotonicity by tailoring the model with specific architectures (Archer & Wang, 1993; Daniels & Velikova, 2010; Kim & Lee, 2024; Milani Fard et al., 2016; You et al., 2017). This approach restricts the hypothesis space of the models, making the implementation challenging and potentially incompatible with the efficient architectures that can yield enhanced performance like residual connections (He et al., 2016).

*2) Monotonicity by regularization.* Another group introduces monotonicity by modifying the training process of general neural networks using heuristical regularization (Gupta et al., 2019a; Liu et al., 2020; Sivaraman et al., 2020). Although this approach offers more flexibility and the monotonicity can be guaranteed through certification (Liu et al., 2020; Sivaraman et al., 2020), it requires empirical tuning of the regularization during training, and the certification process may involve significant computational overhead for large neural networks.

## 1.1 OUR CONTRIBUTION

In this paper, we draw inspirations from conventional constrained optimization and propose a novel, flexible, and adaptive learning framework to induce monotonicity of neural networks with general architectures. Compared to existing methods, our framework exhibits the following three key features.

- **High Flexibility:** Our framework allows to trade off between probability of monotonicity satisfaction and overall prediction performance through a chance constraint, according to the desirability of monotonicity or users' preference;

- **Advanced Capability:** Our framework does not impose any constraints on neural network architectures, thereby inherits many advantages of state-of-the-art deep neural networks, e.g., strong expressivity, easy-to-train properties;

- **Strong Adaptability:** Our framework does not require empirical tuning of the regularization terms prior to training. Instead, it allows adaptive learning without case-by-case pre-processing, making it immediately ready when applying it to a new scenario.

Finally, we evaluate the proposed frameworks on several practical tasks including classification and regression in datasets and controller design in physical systems, highlighting the above features and its competitive performance over state-of-the-art methods in various domains.

## 1.2 RELATED WORK

We further review some concrete examples in each of the aforementioned categories.

**Monotonicity by constructed architectures:** The examples include the non-negative approach (Archer & Wang, 1993), which constrains the weights of monotonic features as positive, Min-Max Network (Sill, 1997; Daniels & Velikova, 2010), which utilizes linear embedding and max-min-pooling layers to impose monotonicity, and Deep Lattice Network (DLN) (You et al., 2017; Milani Fard et al., 2016), which constructs the network with an ensemble of lattice layers. The recent works manipulate the weights of monotonic features. For example, the Lipschitz Monotonic Network (LMN) (Nolte et al., 2022) constrains the norm of weights to attain Lipschitz continuity. In (Runje & Shankaranarayana, 2023), the authors convert updated weights in the monotonic dense layer into absolute values, while the Scalable Monotonic Neural Network (SMNN) (Kim & Lee, 2024) uses the exponentiated units to guarantee monotonicity. These methods require additional operations on the weights and compromise the flexibility, which may result in reduced performance, due to the restricted hypothesis space.

**Monotonicity by regularization:** Various examples in this group modify the training loss by adding a regularization term that penalizes the negative gradients or the negative weights (Sill & Abu-Mostafa, 1996; Gupta et al., 2019b). These methods work for arbitrary architectures but offer no guarantees of monotonicity for the trained models. In (Liu et al., 2020), the authors utilize the piece-wise linear property of ReLU neural networks to verify the monotonicity of the trained models by transforming the verification process into a mixed integer linear programming (MILP) problem. They repetitively increase the regularization strength on negative gradients of the points that are uniformly sampled from the input domain until the model passes the monotonicity certification. However, the certification can be computationally expensive for deep neural networks. Another concurrent method (Sivaraman et al., 2020) utilizes a satisfiability modulo theories (SMT) solver (Barrett & Tinelli, 2018) to find the counterexamples to the monotonicity definition and include the counterexamples in the training data with adjustments to their target values to enforce monotonicity. Monotonicity can be guaranteed by finding the upper and lower envelopes of the model. Similar to (Liu et al., 2020), SMT solver can be computationally expensive as the size of neural networks grows. Furthermore, both (Liu et al., 2020) and (Sivaraman et al., 2020) have been shown to only support the ReLU networks but not general neural networks.

## 2 PRELIMINARIES AND MOTIVATIONS

### 2.1 MONOTONIC FUNCTIONS

Here, we provide the definition of monotonic mapping functions and its equivalent property, including the univariate functions and multivariant functions. The equivalent property of monotonicity is essential for developing our constrained learning algorithm.

**Monotonic Fucntions.** Let $f(x)$ be a univariate continuously differentiable function from an input space $\mathcal{X} \in \mathbb{R}$ to $\mathbb{R}$, where $\mathcal{X} = [l, u]$, with $l < u$. We say that function $f(x)$ is monotonically non-decreasing in $\mathcal{X}$ if $\forall x, x' \in \mathcal{X}, x \leq x'$, the inequality $f(x) \leq f(x')$ is satisfied. Equivalently, we say that function $f(x)$ is monotonically non-decreasing if $\forall x \in \mathcal{X}$, we have $\frac{\partial f(x)}{\partial x} \geq 0$.

**Partial Monotonic Functions.** Let $f(\boldsymbol{x})$ be a multivariate continuous differentiable function from an input space $\boldsymbol{\mathcal{X}} \in \mathbb{R}^d$ to $\mathbb{R}$, where $\boldsymbol{\mathcal{X}} := \times_{1 \leq i \leq d} = \times_{1 \leq i \leq d}[l_i, u_i]$, with $d$ denoting the dimensions of the input space and $l_i < u_i$ for all $1 \leq i \leq d$. Suppose the input vector $\boldsymbol{x} := [x_1, \ldots, x_d] \in \mathbb{R}^d$ can be partitioned into $[\boldsymbol{x}_m, \boldsymbol{x}_{\neg m}]$, where $m$ is a subset of $[1, \ldots, d]$ and $\neg m$ is the associate complement. Similarly, suppose the input space $\boldsymbol{\mathcal{X}}$ can be partitioned into $\boldsymbol{\mathcal{X}}_m$ and its complementary space $\boldsymbol{\mathcal{X}}_{\neg m}$. We say that function $f(\boldsymbol{x})$ is monotonically non-decreasing with respect to $\boldsymbol{x}_m$ if

$$f(\boldsymbol{x}_m, \boldsymbol{x}_{\neg m}) \leq f(\boldsymbol{x}'_m, \boldsymbol{x}_{\neg m}), \ \forall \boldsymbol{x}_m \leq \boldsymbol{x}'_m, \ \forall \boldsymbol{x}_m, \boldsymbol{x}'_m \in \boldsymbol{\mathcal{X}}_m \text{ and } \boldsymbol{x}_{\neg m} \in \boldsymbol{\mathcal{X}}_{\neg m}, \tag{1}$$

where $\leq$ denotes the element-wise inequality for vectors, i.e., $\boldsymbol{x}_m \leq \boldsymbol{x}'_m$ means $x_i \leq x'_i$ for all $i \in m$. Equivalently, the function $f(\boldsymbol{x})$ is monotonically non-decreasing with respect to $\boldsymbol{x}_m$ if $\frac{\partial f(\boldsymbol{x})}{\partial x_i} \geq 0$ holds for all $i \in m$.

### 2.2 MOTIVATIONS

Let $f_{\boldsymbol{\theta}}(\boldsymbol{x})$ denote a neural network model with $\boldsymbol{\theta}$ being the model parameters and $\boldsymbol{x}$ being a data point from the training set $\mathcal{D}$. The model $f_{\boldsymbol{\theta}}$ can be a conventional neural network with commonly used activation functions (e.g., ReLU, Leaky ReLU, Sigmoid, Tanh, etc). Denote by $J\left(f_{\boldsymbol{\theta}}(\boldsymbol{x})\right)$ the typical training loss of a machine learning problem. The conventional training process of the neural network can be formally articulated as solving the unconstrained optimization problem:

$$\min_{\boldsymbol{\theta}} \mathbb{E}_{\boldsymbol{x} \sim \mathcal{D}} J\left(f_{\boldsymbol{\theta}}(\boldsymbol{x})\right), \tag{2}$$

where the expectation $\mathbb{E}$ is with respect to the distribution of $\boldsymbol{x}$, and the stochastic gradient descent (SGD)-based steps are applied to iteratively approach a solution parameter $\boldsymbol{\theta}$. When one seeks to impose monotonically non-decreasing features on the neural network model $f_{\boldsymbol{\theta}}$, the problem formulation becomes:

$$\min_{\boldsymbol{\theta}} \mathbb{E}_{\boldsymbol{x} \sim \mathcal{D}} J\left(f_{\boldsymbol{\theta}}(\boldsymbol{x})\right) \tag{3a}$$

$$\text{s.t.} \quad \frac{\partial f_{\boldsymbol{\theta}}(\boldsymbol{x})}{\partial \boldsymbol{x}_i} \geq 0, \ \forall i \in m, \ \forall \boldsymbol{x} \in \mathcal{D}. \tag{3b}$$

The interpretation of the constrained problem (3) is straightforward from the optimization's perspective. The constraint (3b) confines the feasible solution set, excluding those solutions of (2) that may violate the partial monotonicity requirements. However, problem (3) is not well formulated for directly applying (projected) SGD algorithms to attain a solution due to the existence of constraint (3b). Indeed, it is unclear that how (3b) reflects the constraints on $\boldsymbol{\theta}$ in general. Therefore, the literature has mainly developed two branches to handle it: *monotonicity by constructed architectures* and *monotonicity by regularization*. The former designs neural networks with special architectures so that (3b) is satisfied automatically. This could be understood as the constraints on $\boldsymbol{\theta}$ reflected by (3b) are embedded into the neural network architecture design. The latter instead moves the monotonicity constraint (3b) into the cost function (3a) as a regularization term, and the monotonicity property of neural networks is then promoted through training. Nevertheless, there exist fundamental limits for both of them. For the former, it requires one to modify conventional neural networks with special architectures, which often results in poor expressive capability in practice compared to the conventional neural networks. For the latter, the performance highly depends on the selection and tuning of the regularization term, and there is no systematical method for it.

To overcome these bottlenecks, in the next, we borrow ideas from constrained optimization and propose a novel, flexible, and adaptive learning framework to enforce the monotonicity of neural networks. Specifically, our framework leverages primal-dual algorithms to continuously and adaptively enforce monotonicity for general neural networks, does not impose any constraints on the neural network architectures, and needs no pre-processing such as tuning of the regularization terms.

## 3 CONSTRAINED LEARNING FOR MONOTONIC NETWORKS

Here, we present our proposed framework for training monotonic neural networks. We transform the problem (3) into a chance constrained optimization problem, followed by a stochastic primal-dual learning algorithm to solve it.

### 3.1 REFORMULATION WITH CHANCE CONSTRAINTS

To account for the statistical distribution of constraint (3b) across the entire dataset, we propose a reformulation of problem (3) as below:

$$\min_{\boldsymbol{\theta}} \ \mathbb{E}_{\boldsymbol{x} \sim \mathcal{D}} J\left(f_{\boldsymbol{\theta}}(\boldsymbol{x})\right) \tag{4a}$$

$$\text{s.t.} \ \Pr_{\boldsymbol{x} \sim \mathcal{D}}\left(\frac{\partial f_{\boldsymbol{\theta}}(\boldsymbol{x})}{\partial \boldsymbol{x}_i} \leq 0\right) \leq \alpha, \ \forall i \in m, \tag{4b}$$

where $\Pr_{\boldsymbol{x} \sim \mathcal{D}}$ is a probability operator with respect to the distribution of data samples in the dataset, and $\alpha \in [0, 1]$ is a small number that confines the constraint (3b) satisfaction at least $1 - \alpha$. We note that when $\alpha = 0$, problem (4) is exactly equivalent to the original problem (3). However, when running stochastic primal-dual algorithm to solve (4), the probability operator in (4b) prevents the computation of gradients (i.e., the gradients of (4b) with respect to the parameter $\boldsymbol{\theta}$) that are required by the existing machine learning algorithms. To resolve this issue, notice that

$$\Pr_{\boldsymbol{x} \sim \mathcal{D}}\left(\frac{\partial f_{\boldsymbol{\theta}}(\boldsymbol{x})}{\partial \boldsymbol{x}_i} \leq 0\right) \leq \alpha \iff \mathbb{E}_{\boldsymbol{x} \sim \mathcal{D}}\left[\mathbb{1}\left(0 - \frac{\partial f_{\boldsymbol{\theta}}(\boldsymbol{x})}{\partial \boldsymbol{x}_i}\right)\right] \leq \alpha, \tag{5}$$

for all $i \in m$, where $\mathbb{1}(x)$ is an indicator function that equals 1 if $x \geq 0$ and 0 otherwise. While the condition in the right hand side of (5) is still not continuously differentiable, we show in the next claim a sufficient condition to ensure it yet is continuously differentiable.

**Claim 1.** For any $\boldsymbol{t} = [t_1, \ldots, t_m]^\top \in \mathbb{R}^{|m|}$ with $t_i > 0$ for all $i \in m$, if it holds that $\mathbb{E}_{\boldsymbol{x} \sim \mathcal{D}}\left[\left[\boldsymbol{t} + \boldsymbol{0} - \frac{\partial f_{\boldsymbol{\theta}}(\boldsymbol{x})}{\partial \boldsymbol{x}_m}\right]_+\right] \leq \alpha \boldsymbol{t}$, where $[\cdot]_+$ denotes the projection onto the nonnegative orthant, then $\mathbb{E}_{\boldsymbol{x} \sim \mathcal{D}}\left[\mathbb{1}\left(0 - \frac{\partial f_{\boldsymbol{\theta}}(\boldsymbol{x})}{\partial \boldsymbol{x}_i}\right)\right] \leq \alpha, \forall i \in m$, holds too.

We reason Claim 1 here. Consider some function $g(x)$ of random variable $x$ and the chance constraint $\Pr(g(x) \geq 0) = \mathbb{E}\left[\mathbb{1}\left(g(x)\right)\right] \leq \alpha$, we notice that $\mathbb{1}\left(g(x)\right) \leq \left[1 + g(x)/t\right]_+$ holds for all $g(x)$ and $t > 0$. Therefore, for any $t > 0$ such that $\mathbb{E}\left[\left[1 + g(x)/t\right]_+\right] \leq \alpha$, the constraint $\mathbb{E}\left[\mathbb{1}\left(g(x)\right)\right] \leq \alpha$ holds too. Since $t > 0$, multiplying both sides of $\mathbb{E}\left[\left[1 + g(x)/t\right]_+\right] \leq \alpha$ by $t$ leads to a equivalent condition $\mathbb{E}\left[\left[t + g(x)\right]_+\right] \leq \alpha t$. This leads to Claim 1.

Leveraging Claim 1, by replacing $g(x)$ with $0 - \frac{\partial f_{\boldsymbol{\theta}}(\boldsymbol{x})}{\partial \boldsymbol{x}_i}$ and introducing an auxiliary variable $t_i > 0$ for each $i \in m$, we then have a inner approximation to the chance constraint (5) as $\mathbb{E}\left[\left[t_i + 0 - \frac{\partial f_{\boldsymbol{\theta}}(\boldsymbol{x})}{\partial \boldsymbol{x}_i}\right]_+\right] \leq \alpha t_i, \ \forall i \in m$, which are continuously differentiable. Collecting these constraints over $i \in m$ in a vector form, we have:

$$\min_{\boldsymbol{\theta}, \boldsymbol{t}} \ \mathbb{E}_{\boldsymbol{x} \sim \mathcal{D}} J\left(f_{\boldsymbol{\theta}}(\boldsymbol{x})\right) \tag{6a}$$

$$\text{s.t.} \ \mathbb{E}_{\boldsymbol{x} \sim \mathcal{D}}\left[\left[\boldsymbol{t} + \boldsymbol{0} - \frac{\partial f_{\boldsymbol{\theta}}(\boldsymbol{x})}{\partial \boldsymbol{x}_m}\right]_+\right] \leq \alpha \boldsymbol{t}, \tag{6b}$$

where $\boldsymbol{t} = [t_1, \ldots, t_m] \in \mathbb{R}^{|m|}$. Now, the formulation (6) permits the use of stochastic primal-dual algorithm to optimize the neural network parameter $\boldsymbol{\theta}$ thanks to the continuous differentiability of (6b).

In fact, one can notice that when $\alpha = 0$, problem (6) exactly returns to the original problem (3). This can be understood by checking each data point from the dataset and prohibit any data point from violating the monotonicity requirement, leading to strict monotonicity. On the other hand, selecting a small positive $\alpha$ represents the case where we allow some of the data points violate the monotonicity requirement in a low probability manner. This is particular meaningful in scenarios where monotonicity is acting as a tendency instead of a strict requirement, as in these cases the compromise to monotonicity of a small portion of data points might win significant improvement of overall performance.

**Enforce monotonicity to the whole input space $\mathcal{X}$.** Generally speaking, the dataset $\mathcal{D}$ is a subset of the input domain $\mathcal{X}$ of function $f_{\boldsymbol{\theta}}(\boldsymbol{x})$. To enhance the generalizability of function $f_{\boldsymbol{\theta}}(\boldsymbol{x})$, it is essential to enforce the monotonicity requirement across the entire input domain $\mathcal{X}$ (Liu et al., 2020). Let $\mathrm{Uni}(\mathcal{X})$ denote the uniform distribution on $\mathcal{X}$. In the rest of this paper, we compute the expectation of the chance constraint (6b) over $\mathrm{Uni}(\mathcal{X})$ instead of $\mathcal{D}$, i.e., $\mathbb{E}_{\boldsymbol{z} \sim \mathrm{Uni}(\mathcal{X})} \left[ [\boldsymbol{t} + \boldsymbol{0} - \partial f_{\boldsymbol{\theta}}(\boldsymbol{z})/\partial \boldsymbol{z}_m]_+ \right] \leq \alpha \boldsymbol{t}$, to develop our constrained learning algorithm.

### 3.2 STOCHASTIC PRIMAL-DUAL LEARNING ALGORITHM

Having the problem formulation (6) in mind, here we develop a stochastic primal-dual learning algorithm to approach a solution (Eisen et al., 2019). Let $\boldsymbol{\mu} \in \mathbb{R}^{|m|}$ be the dual variable of the constraint (6b). Consider the Lagrangian function of problem (6):

$$\mathcal{L}(\boldsymbol{\theta}, \boldsymbol{t}, \boldsymbol{\mu}) = \mathbb{E}_{\boldsymbol{x} \sim \mathcal{D}} J\left(f_{\boldsymbol{\theta}}(\boldsymbol{x})\right) + \boldsymbol{\mu}^\top \left( \mathbb{E}_{\boldsymbol{z} \sim \mathrm{Uni}(\mathcal{X})} \left[ \left[ \boldsymbol{t} + \boldsymbol{0} - \frac{\partial f_{\boldsymbol{\theta}}(\boldsymbol{z})}{\partial \boldsymbol{z}_m} \right]_+ \right] - \alpha \boldsymbol{t} \right). \tag{7}$$

The function (7) can be interpreted as a penalized version of problem (6). We penalize the violation of constraint (6b) by a Lagrangian term associated with the dual variable $\boldsymbol{\mu}$. Solving the problem (6) is equivalent to finding a saddle point of the following constraint-free max-min problem:

$$\max_{\boldsymbol{\mu} \geq 0} \min_{\boldsymbol{\theta}, \boldsymbol{t}} \mathcal{L}(\boldsymbol{\theta}, \boldsymbol{t}, \boldsymbol{\mu}). \tag{8}$$

The constraint-free nature of problem (8) enables the use of stochastic gradient-based algorithms in conventional machine learning problems, e.g., SGD algorithm. The minimization over the primal variable $\boldsymbol{\theta}$ is to optimize the performance of neural networks, while the stochastic gradient ascent over the dual variable $\boldsymbol{\mu}$ can be understood as adaptively handling the constraint violations. Specifically, the proposed stochastic primal-dual gradient (SPDG) algorithm reads as follows:

$$\boldsymbol{\theta}^{k+1} = \boldsymbol{\theta}^k - \gamma_{\boldsymbol{\theta}} \nabla_{\boldsymbol{\theta}} \mathcal{L}(\boldsymbol{\theta}^k, \boldsymbol{t}^k, \boldsymbol{\mu}^k), \tag{9a}$$

$$\boldsymbol{t}^{k+1} = \left[ \boldsymbol{t}^k - \gamma_{\boldsymbol{t}} \nabla_{\boldsymbol{t}} \mathcal{L}(\boldsymbol{\theta}^k, \boldsymbol{t}^k, \boldsymbol{\mu}^k) \right]_+, \tag{9b}$$

$$\boldsymbol{\mu}^{k+1} = \left[ \boldsymbol{\mu}^k + \gamma_{\boldsymbol{\mu}} \nabla_{\boldsymbol{\mu}} \mathcal{L}(\boldsymbol{\theta}^{k+1}, \boldsymbol{t}^{k+1}, \boldsymbol{\mu}^k) \right]_+, \tag{9c}$$

where $(\gamma_{\boldsymbol{\theta}}, \gamma_{\boldsymbol{t}})$ and $\gamma_{\boldsymbol{\mu}}$ are positive learning rates for the primal and dual variables, respectively. To align with the prevalent minibatch training style, we suppose the SPDG algorithm is iteratively performed over batched samples $\{\boldsymbol{x}^s\}_{s=1}^S$ from $\mathcal{D}$ and $\{\boldsymbol{z}^n\}_{n=1}^N$ from $\mathrm{Uni}(\mathcal{X})$. Using the rule in stochastic gradient descent, the gradient term in the primal update (9a) can be computed as:

$$\nabla_{\boldsymbol{\theta}} \mathcal{L} = \frac{1}{S} \sum_{s=1}^S \nabla_{\boldsymbol{\theta}} J\left(f_{\boldsymbol{\theta}}(\boldsymbol{x}^s)\right) + \frac{1}{N} \sum_{n=1}^N \boldsymbol{\mu}^\top \nabla_{\boldsymbol{\theta}} \left( \left[ \boldsymbol{t} + \boldsymbol{0} - \frac{\partial f_{\boldsymbol{\theta}}(\boldsymbol{z}^n)}{\partial \boldsymbol{z}_m} \right]_+ - \alpha \boldsymbol{t} \right). \tag{10}$$

Indeed, the gradient (10) can be readily obtained through the backpropagation of the neural network by letting $\mathcal{L}$ as the loss function. The first term in (10) is the descent direction of the original problem (2), while the second term rectifies the descent direction considering the constraint (6b). As for the gradient terms in (9b) and (9c), they can be computed as:

$$\nabla_{\boldsymbol{t}} \mathcal{L} = \frac{1}{N} \sum_{n=1}^N \mathbb{1}\left( \boldsymbol{t} - \frac{\partial f_{\boldsymbol{\theta}}(\boldsymbol{z})}{\partial \boldsymbol{z}_m} \right) \cdot \boldsymbol{\mu} - \alpha \boldsymbol{\mu}, \text{ and } \nabla_{\boldsymbol{\mu}} \mathcal{L} = \frac{1}{N} \sum_{n=1}^N \left[ \boldsymbol{t} - \frac{\partial f_{\boldsymbol{\theta}}(\boldsymbol{z})}{\partial \boldsymbol{z}_m} \right]_+ - \alpha \boldsymbol{t}. \tag{11}$$

We summarize the proposed stochastic primal-dual learning algorithm for training monotonic neural networks in Algorithm 1. A key advantage of the proposed method is that we do not impose any

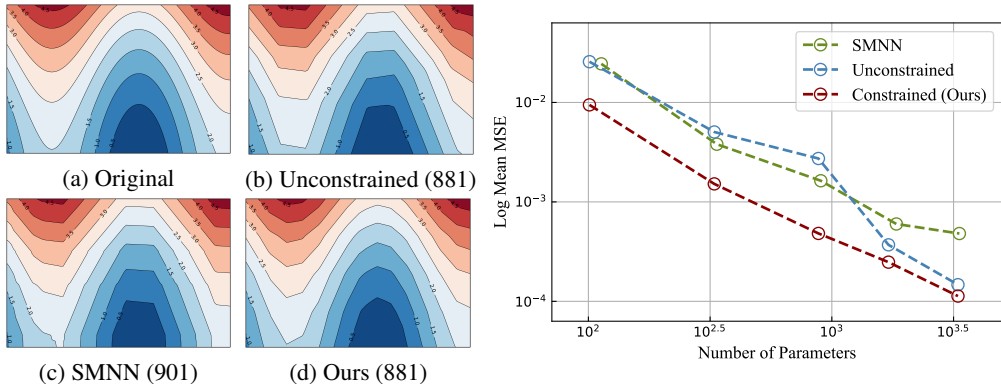

Figure 1: Comparison of unconstrained network, SMNN (Kim & Lee, 2024), and our method on a 2D example from (Liu et al., 2020), i.e., fitting $f(x, y) = a \sin(x/25\pi) + b(x - 0.5)^3 + c \exp(y) + y^2$, $a, b, c \in \{0.3, 0.6, 1.0\}$, $x, y \in [0, 1]$. **Left:** Contour plots when $a, b, c = 1.0$ with the number in the parenthesis denoting the number of parameters. Our method shows the best fitting result. **Right:** Evaluation of three methods on all combinations of $a, b, c$ with increasing network size. The lines denote the average MSE of 27 runs. Our method outperforms the other two methods.

constraints on the neural network architectures. This is desirable as it can take advantage of classical neural networks, which often have stronger expressivity and are easier to train. Observing the dual update (9c) and the gradient $\nabla_{\boldsymbol{\mu}}\mathcal{L}$, we can find that the dual variable $\boldsymbol{\mu}$ will automatically modulate the penalty strength based on the degree of constraint satisfaction. This adjustment occurs seamlessly, obviating the manual tuning of regularization like (Liu et al., 2020). In this sense, we can interpret Algorithm 1 as an adaptive regularization approach. In practice, one may also consider to fix the auxiliary variable $\boldsymbol{t}$ at a small positive constant vector to further ease the training.

---

**Algorithm 1:** Stochastic Primal-Dual Learning Algorithm

---

Randomly initialize neural network $f_{\boldsymbol{\theta}}(\boldsymbol{x})$, initialize $\boldsymbol{t} = \boldsymbol{0}$, $\boldsymbol{\mu} = \boldsymbol{0}$

**Input:** Dataset $\mathcal{D}$, Input domain $\mathcal{X}$

**for** *epoch*$= 0, 1, 2, \dots$ **do**

  Randomly draw samples $\{\boldsymbol{x^s}\}_{s=1}^S$ from $\mathcal{D}$ and $\{\boldsymbol{z}^n\}_{n=1}^N$ from $\mathrm{Uni}(\mathcal{X})$

  Observe the derivatives $\frac{\partial f_{\boldsymbol{\theta}}(\boldsymbol{z})}{\partial \boldsymbol{z}_m}|_{\boldsymbol{z}=\boldsymbol{z}^n}$ of the neural network

  Use backpropagation to compute $\nabla_{\boldsymbol{\theta}}\mathcal{L}$ via (10), and update the neural network parameter:

$$\boldsymbol{\theta}^{k+1} = \boldsymbol{\theta}^k - \gamma_{\boldsymbol{\theta}}\nabla_{\boldsymbol{\theta}}\mathcal{L}(\boldsymbol{\theta}^k, \boldsymbol{t}^k, \boldsymbol{\mu}^k)$$

  Compute the gradients $\nabla_{\boldsymbol{t}}\mathcal{L}$ and $\nabla_{\boldsymbol{\mu}}\mathcal{L}$ by (11). Update the auxiliary and dual variables:

$$\boldsymbol{t}^{k+1} = \left[\boldsymbol{t}^k - \gamma_{\boldsymbol{t}}\nabla_{\boldsymbol{t}}\mathcal{L}(\boldsymbol{\theta}^k, \boldsymbol{t}^k, \boldsymbol{\mu}^k)\right]_+, \text{ and } \boldsymbol{\mu}^{k+1} = \left[\boldsymbol{\mu}^k + \gamma_{\boldsymbol{\mu}}\nabla_{\boldsymbol{\mu}}\mathcal{L}(\boldsymbol{\theta}^{k+1}, \boldsymbol{t}^{k+1}, \boldsymbol{\mu}^k)\right]_+$$

**end**

---

## 4 EXPERIMENTS

In this section, we demonstrate the effectiveness of our proposed method in various practical tasks. We first conduct experiments on a 2D example from (Liu et al., 2020), and Figure 1 shows the results. We then conduct experiments on public datasets from (Liu et al., 2020; Sivaraman et al., 2020; Nolte et al., 2022; Runje & Shankaranarayana, 2023; Kim & Lee, 2024) and compare them with state-of-the-art methods. The results show that our method can achieve higher accuracies while using fewer model parameters. Finally, we conduct experiments on a real-world safe-critical frequency control system from (Cui et al., 2023a). Our proposed method also shows improved performance.

### 4.1 EXPERIMENTS ON PUBLIC DATASETS

Table 1: Comparison of our method with other methods presented in (Liu et al., 2020; Nolte et al., 2022; Runje & Shankaranarayana, 2023; Kim & Lee, 2024).

| Method | COMPAS | | Blog Feedback | | Loan Defaulter | |
|---|---|---|---|---|---|---|
| | Parameters | Test Acc ↑ | Parameters | RMSE ↓ | Parameters | Test Acc ↑ |
| Isotonic | N.A. | 67.6% | N.A. | 0.203 | N.A. | 62.1% |
| XGBoost | N.A. | $68.5\% \pm 0.1\%$ | N.A. | $0.176 \pm 0.005$ | N.A. | $63.7\% \pm 0.1\%$ |
| Crystal | 25840 | $66.3\% \pm 0.1\%$ | 15840 | $0.164 \pm 0.002$ | 16940 | $65.0\% \pm 0.1\%$ |
| DLN | 31403 | $67.9\% \pm 0.3\%$ | 27903 | $0.161 \pm 0.001$ | 29949 | $65.1\% \pm 0.2\%$ |
| Min-Max Net | 42000 | $67.8\% \pm 0.1\%$ | 27700 | $0.163 \pm 0.001$ | 29000 | $64.9\% \pm 0.1\%$ |
| Non-Neg-DNN | 23112 | $67.3\% \pm 0.9\%$ | 8492 | $0.168 \pm 0.001$ | 8502 | $65.1\% \pm 0.1\%$ |
| Certified MNN | 23112 | $68.8\% \pm 0.2\%$ | 8492 | $0.158 \pm 0.001$ | 8502 | $65.2\% \pm 0.1\%$ |
| LMN | 37 | $69.3\% \pm 0.1\%$ | 2225 | $0.160 \pm 0.001$ | **753** | $\mathbf{65.4\% \pm 0.0\%}$ |
| Constrained MNN | 2317 | $69.2\% \pm 0.2\%$ | 1101 | $0.156 \pm 0.001$ | 177 | $65.3\% \pm 0.1\%$ |
| SMNN | 2657 | $69.3\% \pm 0.9\%$ | **1421** | $\mathbf{0.150 \pm 0.001}$ | 501 | $65.0\% \pm 0.1\%$ |
| Ours | **2069** | $\mathbf{69.4\% \pm 0.1\%}$ | 847 | $0.151 \pm 0.001$ | **673** | $\mathbf{65.4\% \pm 0.0\%}$ |

We conduct experiments on five publicly available datasets that were used in previous studies (Liu et al., 2020; Sivaraman et al., 2020; Nolte et al., 2022; Runje & Shankaranarayana, 2023; Kim & Lee, 2024): COMPAS, Blog feedback, Loan defaulter, Auto MPG, and Heart disease. COMPAS is a classification dataset with 13 features of which 4 are monotonic. Blog feedback is a regression dataset with 276 features of which 8 are monotonic. Loan defaulter is a classification dataset with 28 features of which 5 are monotonic. Auto MPG is a regression dataset with 7 features of which 3 are monotonic and Heart disease is a classification dataset with 13 features of which 2 are monotonic. Appendix A provides more details.

Table 2: Comparison of our method with other methods presented in (Sivaraman et al., 2020; Nolte et al., 2022; Runje & Shankaranarayana, 2023; Kim & Lee, 2024).

| Method | Auto MPG | Heart Disease |
|---|---|---|
| | MSE ↓ | Test Acc ↑ |
| Min-Max Net | $10.14 \pm 1.54$ | $0.75 \pm 0.04$ |
| DLN | $13.34 \pm 2.42$ | $0.86 \pm 0.02$ |
| COMET | $8.81 \pm 1.81$ | $0.86 \pm 0.03$ |
| LMN | $7.58 \pm 1.20$ | $0.90 \pm 0.02$ |
| Constrained MNN | $8.37 \pm 0.08$ | $0.89 \pm 0.00$ |
| SMNN | $7.44 \pm 1.20$ | $0.88 \pm 0.04$ |
| Ours | $\mathbf{5.82 \pm 0.26}$ | $\mathbf{0.92 \pm 0.14}$ |

For the classification tasks, we use cross-entropy loss for training, and for the regression tasks, we use mean-square-error loss. For all the tasks, we utilize the Adam optimizer to train the primal variable (model parameter) $\theta$ and update the dual variable using the rule (9c). We initialize the dual variable $\mu = 0$, and fix the auxiliary variable $t = 1 \times 10^{-4}$. The chance constraint coefficient $\alpha$ is set as 0.1 for all experiments. The more detailed configurations of the experiments are available in Appendix A. The proposed method is compared with the benchmarking methods in (Liu et al., 2020; Runje & Shankaranarayana, 2023; Kim & Lee, 2024) and other methods described in them that can generate partial monotonic models. The methods include Isotonic Regression (Kalai & Sastry, 2009), XGBoost (Chen et al., 2015), Crystal (Milani Fard et al., 2016), Deep Lattice Network (DLN) (You et al., 2017), Min-Max Net (Daniels & Velikova, 2010), Non-Neg-DNN (Archer & Wang, 1993), Certified Monotonic Neural Network (Certified MNN) (Liu et al., 2020), Counterexample-Guided Learning (COMET) (Sivaraman et al., 2020), Lipschitz Monotonic Network (LMN) (Nolte et al., 2022), Constrained MNN (Runje & Shankaranarayana, 2023), and Scalable Monotonic Neural Network (SMNN) (Kim & Lee, 2024).

Tables 1 and 2 present the experimental results on the datasets above. We evaluate the models with metrics such as test accuracy for classification and mean squared error and root mean squared error (MSE and RMSE) for regression, and compare the model complexity (i.e., number of parameters) if available. We run the experiments ten times per dataset after finding the optimal hyperparameters and report the mean and standard deviation of the best five results, which is aligned with previous studies (Runje & Shankaranarayana, 2023; Kim & Lee, 2024). Table 1 shows that our method can attain the top positions in COMPAS and Loan Defaulter datasets. Although LMN also secures the top position in Loan Defaulter, it needs more model parameters. As for the Blog Feedback dataset where the RMSE of our method is slightly larger than SMNN, our method uses the smallest model sizes while achieving comparable performance. Table 2 also demonstrates the superior performance of our method in Auto MPG and Heart Disease datasets. We can see that our method outperforms all the

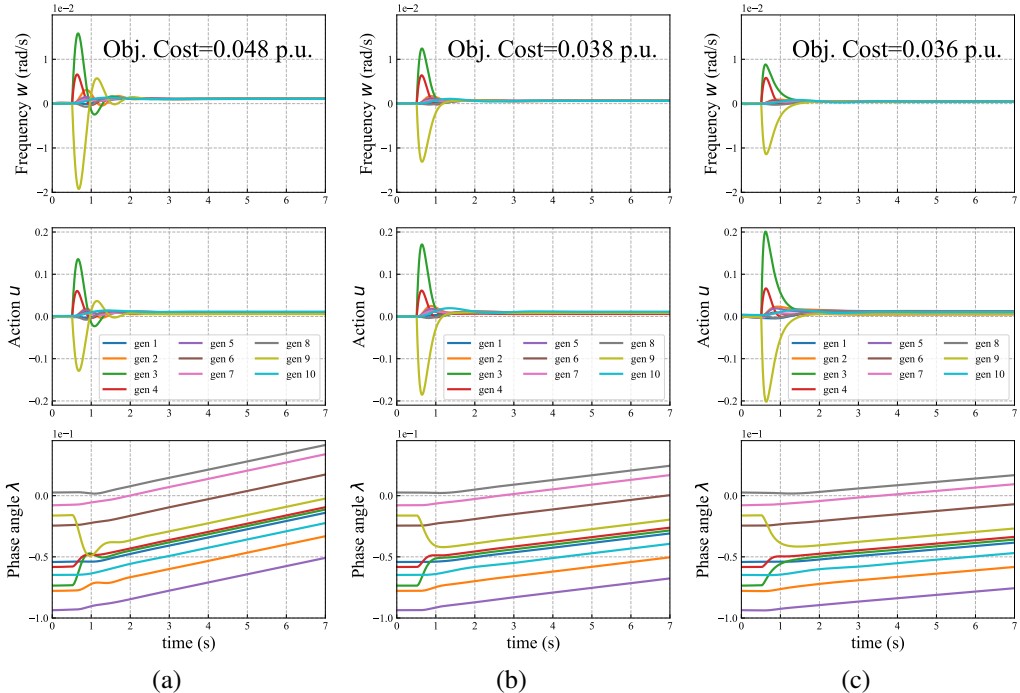

Figure 2: Comparison of the control trajectories and objective costs on a real-world frequency control system. The trajectories include frequency deviation $w$, control action $u_\theta$ and phase angle $\lambda$ on selected 10 buses. The numbers in the figures denote the objective cost (12a). The monotonic controllers are modeled by three methods: (a) the SMNN from (Kim & Lee, 2024); (b) the monotonic SNN in the original paper (Cui et al., 2023a); (c) our constrained learning NN.

benchmarking methods with significant improvements. It is worth noting that our method, designed with general network architectures, is easier to train than customized architecture methods such as the recent state-of-the-art SMNN. Compared to Certified MNN, which requires manually increasing the coefficient of regularization that may cause training failures with excessive penalties, our method does not encounter any training failure cases.

## 4.2 EXPERIMENTS ON FREQUENCY CONTROL

The previous works (Liu et al., 2020; Runje & Shankaranarayana, 2023; Kim & Lee, 2024) predominantly focused on supervised learning tasks like classification and regression as discussed in subsection 4.1, leaving uncertainties about their monotonicity frameworks in unsupervised learning domains. In this subsection, we extend the experiments to an unsupervised learning task, namely, a task that is trained by reinforcement learning algorithms, to further assess the performance of the proposed method. We consider an optimal frequency control problem in a real-world power system from (Cui et al., 2023a). The power network comprises 39 buses with 10 of them integrated with inverter-connected resources. These resources, equipped with controllers, are deployed in the power system to provide frequency regulation to resist power disturbances. As the power network connects numerous end users and large amounts of vulnerable devices, the stability of the closed-loop system built on the controllers is essential. This necessitates that the control actions exhibit monotonicity concerning the control inputs, which is also analyzed using Lyapunov stability theory (Cui et al., 2023a). The optimal controllers are obtained by solving the following optimization problem:

$$\min_{\theta} \sum_{i}^{n} \left( \|\boldsymbol{\omega_i}\|_{\infty} + \eta C(\boldsymbol{u_{\theta_i}}) \right) \tag{12a}$$

$$\text{s.t. } \left( \lambda_i(k+1), \omega_i(k+1) \right) = F \left( \lambda_i(k), \omega_i(k) \right) + G \left( u_{\theta_i}(\omega_i(k)) \right), \tag{12b}$$

$$u_{\theta_i}(\cdot) \text{ is monotonically increasing.} \tag{12c}$$

where $\omega_i$ and $\lambda_i$ are the frequency deviation and phase angle of bus $i$, respectively, and $u_{\theta_i}(\cdot)$ is a controller that is typically parametrized by a neural network. The bold symbols $\boldsymbol{\omega_i} = (\omega_i(0), \ldots, \omega_i(K))$

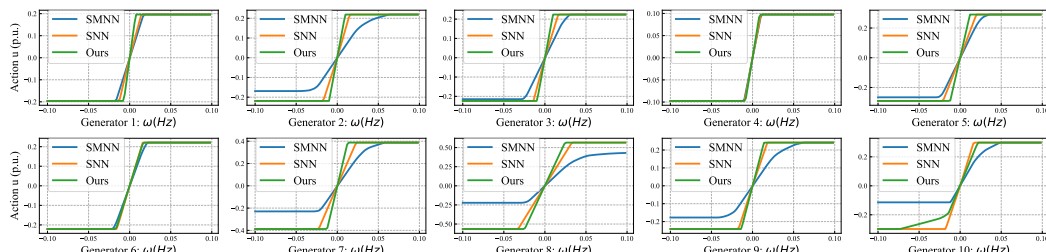

Figure 3: The input-output plots of the learned controllers modeled by SMNN (Kim & Lee, 2024), monotonic SNN (Cui et al., 2023a), and our constrained learning. Each controller is confined by a feasible region $\underline{u}_i \leq u_{\theta_i} \leq \overline{u}_i$. All three methods can learn monotonic controllers. However, the SMNN controllers truncate the default feasible region of generators $2, 5, 7, 8, 9, 10$.

and $\boldsymbol{u}_{\boldsymbol{\theta_i}} = (u_{\theta_i}(\omega_i(0)), \ldots, u_{\theta_i}(\omega_i(K)))$ represent the control trajectories over $K$ steps. Equation (12b) describes the state transition of the power system, framing (12) as a quintessential optimal control problem. In this experiment, we utilize the reinforce-style algorithm developed in (Cui et al., 2023a) to train the controllers but impose the monotonicity constraint (12c) into controller $u_{\theta_i}(\cdot)$ through three different methods: monotonic single-layer neural network (SNN) in the original paper (Cui et al., 2023a), the recent state-of-the-art SMNN (Kim & Lee, 2024), and our constrained learning NN. We set the coefficient $\eta = 4 \times 10^{-4}$, and train the three models with Adam optimizer in identical environments (i.e., the power system under the same set of power disturbances). As for our proposed method, we initialize the dual variable $\boldsymbol{\mu} = 0$ and fix the auxiliary variable $\boldsymbol{t} = 1 \times 10^{-4}$. The chance constraint coefficient $\alpha$ is set as $0.1$ and the learning rate $\gamma_{\boldsymbol{\mu}}$ for the dual variable is $10$. Appendix B provides more comprehensive descriptions of problem (12) and the corresponding configurations.

Figure 2 compares the trajectories of frequency deviation $\boldsymbol{\omega}$, control action $\boldsymbol{u_\theta}$ and phase angle $\boldsymbol{\lambda}$ in the buses equipped with controllers after an identical power disturbance was injected to the power system. We evaluate the models using the objective cost (12a). We conduct five independent runs on each model and plot the best results in Figure 2. It shows that all three models can stabilize the power system after a power disturbance occurs at time$= 0.5$s. Our method outperforms the other two methods by achieving the lowest objective cost, exhibiting $25.0\%$ improvement over SMNN and $5.3\%$ improvement over monotonic SNN, respectively. This is mainly because our method allows the use of general neural networks that are easier to train, which may help the algorithm attain better controllers. From the upper three plots in Figure 2, we see that the frequency deviation $\boldsymbol{\omega}$ achieved by our method is notably smaller than that of SMNN and monotonic SNN. The middle three plots explain the reason since our method adopts the more aggressive control actions to promptly dampen the frequency deviations. The bottom three plots further demonstrate that our method maintains a more consistent alteration in phase angles, attributed to the smaller frequency deviations achieved.

Figure 3 compares the input-output plots of the learned controllers. It shows that all three methods can produce monotonic controllers. However, a closer inspection reveals that the output regions of SMNN controllers in generators $2, 5, 7, 8, 9, 10$ are comparatively constrained, falling short of the default feasible regions. Such limitations are undesirable in practical control and might consequently result in inferior performances. Although monotonic SNN and our method yield similar plots from generator 1 to 9, our method excels in learning a nonlinear controller in generator 10. This superiority is also verified by the results in Figure 2, where our method achieves the lowest objective cost.

## 5 CONCLUSIONS AND FUTURE WORK

In this paper, we have proposed a novel, flexible, and adaptive learning framework to enforce monotonicity of neural networks with general architectures. Our framework relies on a primal-dual learning algorithm with only small extra computations to continuously and adaptively enforce the monotonicity until the constraint is satisfied. It does not impose any constraints on the neural network architectures nor needs case-by-case pre-processing such as tuning of the regularization. Experiments on various practical tasks show that our method achieves competitive performance compared to recent state-of-the-art methods. In the future work, we will extend the idea presented in this paper to train neural networks with general inequality constraints, and consider its applications in other control problems of safety-critical systems, e.g. robotic systems and sustainable energy systems.

## ETHICS STATEMENT

In this paper, we propose a framework to train monotonic neural networks. Our method can be utilized to address safety issues when applying machine learning in physical systems. Therefore, our framework is a defensive method and our work does not discover any new threat. Our research also does not include any human subjects. Accordingly, this paper does not raise ethical issues.

## REPRODUCIBILITY STATEMENT

We are committed to make to all aspects of our work open-source, and provide comprehensive instructions for guaranteeing reproducibility. All essential details necessary for reproducing our experiments can be founded in Section 4 and Appendices A, B. The datasets, algorithms, and pretrained models of our method are provided in supplementary materials.

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

## A  DETAILS OF EXPERIMENT 4.1

### A.1  DETAILS OF DATASETS

We provide a detailed description of the five real-world datasets in Table 3. The datasets are chosen from (Liu et al., 2020) and (Sivaraman et al., 2020), which are used in previous studies (Runje & Shankaranarayana, 2023; Nolte et al., 2022; Kim & Lee, 2024) for evaluating model performance. Table 3 includes the tasks of the datasets, number of features, number of monotonic features, the size of the training set and the testing set. Following (Kim & Lee, 2024), we multiply the monotonically non-increasing features by $-1$ to make them non-decreasing. All the inputs are normalized into the range $[0, 1]$.

Table 3: Summary of the real-world datasets.

| Dataset | Task | #Features | #Monotonic features | #Train | #Test |
|---|---|---|---|---|---|
| COMPAS | Classification | 13 | 4 | 4937 | 1235 |
| Blog Feedback | Regression | 276 | 8 | 47302 | 6968 |
| Loan Defaulter | Classification | 28 | 5 | 418697 | 70212 |
| Auto MPG | Regression | 7 | 3 | 313 | 79 |
| Heart Disease | Classification | 13 | 2 | 242 | 61 |

**COMPAS (Angwin et al., 2016):**  COMPAS is a classification dataset containing the criminal records of 6127 individuals arrested in Florida. The goal of this task is to predict whether an arrested individual will commit a crime or not within two years. We use 13 input features for prediction and the classification is based on the predicted risk scores. In the dataset, the risk score is expected to be monotonically non-decreasing in 4 features: *number of prior adult convictions*, *number of juvenile felony*, *number of juvenile misdemeanor*, and *number of other convictions*.

**Blog Feedback (Buza, 2013):**  Blog feedback is a regression dataset comprising 54270 samples from blog posts. The goal of this task is to predict the number of comments in the upcoming 24 hours based on 276 features. Among all the features, 8 of them are expected to be monotonically non-decreasing with the prediction, which are (A51, A52, A53, A54, A56, A57, A58, A59).

**Loan Defaulter:**  The Loan Defaulter[1] dataset contains complete loan data for all loans issued between 2007 and 2015 of several banks. The dataset contains 488909 data samples and each sample has 28 features including the current loan status, the latest payment, and other additional information. The goal of this task is to predict the possible loan defaulters. The prediction is expected to be monotonically non-decreasing in *number of public record bankruptcies*, *debt-to-income ratio* and monotonically non-increasing in *credit score*, *length of employment*, and *annual income*.

**Auto MPG:**  Auto MPG[2] is a regression dataset containing 392 data samples. The goal of the task is to predict city-cycle fuel consumption in miles per gallon (MPG) based on 7 features. Among them, *weights*, *displacement*, and *horse power* are monotonically non-increasing features with the prediction.

**Heart Disease:**  Heart Disease[3] is a classification dataset containing 303 data samples. The task is to predict the presence of heart disease for a person based on 13 features. The risk of heart disease should be monotonically increasing with respect to *trestbp (resting blood pressure)* and *chol (cholesterol level)*.

### A.2  DETAILS OF CONFIGURATIONS

We build the neural network models using a simple MLP structure with ReLU as the activation function. The model consists of three or four layers depending on specific tasks. For the hyperparameters

---

[1] https://www.kaggle.com/wendykan/lenging-club-loan-data
[2] https://archive.ics.uci.edu/dataset/9/auto+mpg
[3] https://archive.ics.uci.edu/dataset/45/heart+disease

in the proposed constrained learning algorithm, we initialize the dual variables $\boldsymbol{\mu} = \mathbf{0}$ and fix the auxiliary variable $\boldsymbol{t} = 1 \times 10^{-4}$. We set the learning rate for the dual variable as $\gamma_{\boldsymbol{\mu}} = 10$ and set the chance constraint coefficient as $\alpha = 0.1$ for all experiments. The number of data points sampled from $\mathrm{Uni}(\boldsymbol{\mathcal{X}})$ is set as $N = 128$. We train the model for 1000 epochs on each of the datasets. Table 4 summarizes the other hyperparameters such as numbers of parameters, network structures, batch sizes, and learning rates for the models in the five datasets.

Table 4: Hyperparameters of the networks on the datasets. For network structure, the two numbers in parenthesis mean that the input layer is partitioned into two parts. The first number denotes the layer size corresponding to the monotonic features, while the second number denotes the layer size of non-monotonic features.

| Dataset | #Parameters | Network structure | Batch size | Learning rate |
|---|---|---|---|---|
| COMPAS | 2069 | $(4, 32) - 32 - 16 - 1$ | 256 | $5 \times 10^{-4}$ |
| Blog Feedback | 847 | $(1, 3) - 3 - 1$ | 256 | $5 \times 10^{-4}$ |
| Loan Defaulter | 673 | $(8, 16) - 8 - 4 - 1$ | 512 | $5 \times 10^{-4}$ |
| Auto MPG | 417 | $(16, 16) - 8 - 1$ | 128 | $5 \times 10^{-3}$ |
| Heart Disease | 1353 | $(8, 32) - 16 - 16 - 1$ | 128 | $2 \times 10^{-4}$ |

# B    DETAILS OF EXPERIMENT 4.2

## B.1    DESCRIPTION OF OPTIMAL FREQUENCY CONTROL TASK

In this subsection, we provide a detailed description of problem (12). The formulation of the optimal frequency control task can be formally written as the optimization problem below (Cui et al., 2023a):

$$\min_{\theta} \sum_{i}^{n} \left( \|\boldsymbol{\omega_i}\|_{\infty} + \eta C(\boldsymbol{u_{\theta_i}}) \right) \tag{13a}$$

$$\text{s.t. } \lambda_i(k) = \lambda_i(k-1) + \omega_i(k-1)\Delta t, \tag{13b}$$

$$\omega_i(k) = -\frac{\Delta t}{M_i} \sum_{j=1}^{|\mathcal{B}|} B_{ij} \sin\left(\lambda_{ij}(k-1)\right) + \frac{\Delta t}{M_i} p_{m,i}$$

$$+ \left(1 - \frac{D_i \Delta t}{M_i}\right) \omega_i(k-1) - \frac{\Delta t}{M_i} u_{\theta_i}(\omega_i(k-1)), \tag{13c}$$

$$\underline{u}_i \leq u_{\theta_i}(\omega_i(k)) \leq \overline{u}_i, \tag{13d}$$

$$\omega_i(k) u_{\theta_i}(\omega_i(k)) \geq 0, \tag{13e}$$

$$u_{\theta_i}(\cdot) \text{ is monotonically increasing.} \tag{13f}$$

where equations (13b) and (13c) are the system transition functions, which are compactly written as $(\lambda_i(k+1), \omega_i(k+1)) = F(\lambda_i(k), \omega_i(k)) + G(u_{\theta_i}(\omega_i(k)))$ in problem (12). We can interpret equations (13b) and (13c) as a deterministic environment[4] and $u_{\theta_i}(\cdot)$ is the policy of a smart agent, which aims to minimize the cost $\sum_{i}^{n} (\|\boldsymbol{\omega_i}\|_{\infty} + \eta C(\boldsymbol{u_{\theta_i}}))$ under the random power disturbance $p_{m,i}$. The infinity norm over the control trajectory $\boldsymbol{\omega_i} = (\omega_i(0), \dots, \omega_i(K))$ is defined as $\|\boldsymbol{\omega_i}\|_{\infty} = \max_{k=0,\dots,K} |\omega_i(k)|$. The explanation of other symbols in (13b) and (13c) can be found in their paper (Cui et al., 2023a). Inequality (13d) confines the feasible region of controller $i$, which can be realized by clipping the output of the controller. Each controller is single-input single-output, and the system has 10 controllers in total. The constraints (13e) and (13f) describe the expected property of the designed controllers to guarantee the exponential stability of the system. We use the RNN-based reinforcement learning algorithm developed in (Cui et al., 2023a) to train the controllers based on three different models of $u_{\theta_i}(\cdot)$. The evaluation of the model performance is simple, as the smaller objective cost implies the better performance of the models.

---

[4](Cui et al., 2023a): https://github.com/Wenqi-Cui/RNN-RL-Frequency-Lyapunov

## B.2 DETAILS OF EXPERIMENTAL CONFIGURATIONS

For our proposed method, we use the ReLU neural network to build the controllers. Each controller $u_{\theta_i}(\cdot)$ consists of three layers and the network structure is $32 - 32 - 1$ for $i = 1, \ldots, 10$. For fair comparisons, we use the same network structure (i.e., $32 - 32 - 1$) for SMNN and use the default structure for monotonic SNN in the original paper. For all three methods, we set the length of an episode as $K = 200$ and the batch size as $600$. We use the Adam optimizer with the learning rate $5 \times 10^{-3}$ to train the controllers for $600$ episodes. In the testing stage, we evaluate the controllers modeled by three different methods using the same power disturbance $p_{m,i}$ for fairness.

