# OpenReview forum: "Shaping Monotonic Neural Networks with Constrained Learning"
_ICLR.cc/2026/Conference — Submitted to ICLR 2026_

### Official Review · Reviewer_8q32 · 2025-10-25

**Soundness:** 2
**Presentation:** 3
**Contribution:** 1
**Rating:** 2
**Confidence:** 3

**Summary:**

The paper presents a new method for constructing monotonic neural networks, applicable to any neural network architecture. Instead of modifying the network's architecture to enforce monotonicity, as is common in previous work, the proposed approach introduces a primal-dual learning algorithm inspired by constrained optimization. The core idea is to reformulate the training task as a constrained optimization problem, where the partial derivatives of the output with respect to selected monotonic input features must be non-negative. To make this constraint tractable and differentiable, it is relaxed into a chance constraint and further approximated by a differentiable surrogate.

**Strengths:**

The paper is well-written and clearly presented. It introduces a novel method for constructing monotonic neural networks. One of the key strengths of this approach is its compatibility with any type of neural network architecture. The proposed technique is both conceptually promising.

**Weaknesses:**

The central issue with this work is that, although the authors claim to introduce a novel method for enforcing monotonicity, the approach provides no formal guarantees of monotonic behavior. The use of chance constraints inherently allows a fraction of violations, as the algorithm permits up to an α proportion of samples to breach the monotonicity condition. Consequently, the resulting model may still produce non-monotonic outputs for certain inputs.

Furthermore, to impose monotonicity across the input domain X, the method samples points uniformly from X and enforces non-negative gradients at those locations. This strategy assumes that a representative uniform distribution over X can be defined and sampled, which is often unrealistic. In high-dimensional or complex domains, uniformly sampling a limited number of points is unlikely to capture all relevant regions, meaning monotonicity may still be violated in unsampled areas.
These limitations make the evaluation procedure, comparing with other methods, fundamentally unreliable. Competing methods guarantee monotonicity throughout the input space, whereas the proposed method enforces offers weak or no guarantees. As a result, the reported comparisons are unreliable, since it remains unclear whether the proposed approach truly preserves monotonicity. In fact, the model could violate the constraint extensively while still exhibiting strong predictive performance. Lastly, the empirical results fail to show any significant improvement over existing methods.

**Questions:**

Could you provide an analysis of the constraint violations? Can you consider alternative sampling strategies, such as drawing samples from distributions that better align with the data manifold?

---

### Official Review · Reviewer_e4aA · 2025-10-31

**Soundness:** 1
**Presentation:** 4
**Contribution:** 1
**Rating:** 2
**Confidence:** 5

**Summary:**

This paper proposes a framework for training neural networks that are monotonic. Instead of enforcing monotonicity by design, the authors treat it as a constraint in the training problem. The proposed solution uses a stochastic primal-dual optimization method, converting the monotonicity requirement into a chance constraint.
The approach is evaluated aganist previous literature on a standard set of benchmarks.

**Strengths:**

The paper is very clearly structured and easy to follow.

**Weaknesses:**

The original contribution of the paper seems a bit overstated and unclear. The first two advantages described in Section 1.1 are shared by previous literature and are not unique of the proposed method. As for the third one, the proposed method has a single hyperparamer that regulates the trade-off between monotonicity and task accuracy, which is the same as in previous literature.

The experimental section is very lacking. The authors do not show empirically hat the method is able to achieve monotonicity in practice. One of the advantages stated is the ability to regulate the percentage of monotonicity. This aspect has not been investigated empirically.
Finally, the proposed method is compared mostly to approaches that provide monotonicity guarantees. This comparison is unfair since it has not been shown that the network obtained using the proposed method is 100% monotonic.

**Questions:**

The percentage of monotonicity achieved is not shown. Why is this the case? Is it always 100%? Does this change for different values of alpha?

---

### Official Review · Reviewer_5PMn · 2025-10-31

**Soundness:** 2
**Presentation:** 2
**Contribution:** 2
**Rating:** 2
**Confidence:** 4

**Summary:**

This paper proposes an architecture-agnostic framework to train neural networks whose outputs are monotonic with respect to selected inputs by reformulating monotonicity as a chance constraint and solving it via a stochastic primal–dual (SPDG) method, using a violation tolerance $ \alpha $ to directly trade off monotonicity and predictive performance. A dual variable automatically adjusts penalties based on the degree of violation, and constraints are evaluated using uniform sampling over the full input domain to promote monotonic behavior beyond the training data. Across multiple public datasets (classification and regression) and a real-world power-system frequency-control task, the method achieves competitive or superior performance to existing monotonic NNs (e.g., DLN, LMN, SMNN) while remaining parameter-efficient.

**Strengths:**

- **S1**. Novelty of the primal–dual formulation for enforcing partial monotonicity: it recasts monotonicity as a chance constraint and introduces dual penalties, integrating a differentiable surrogate constraint into standard training.

- **S2**. Architecture-agnostic control of the fit–monotonicity trade-off without bespoke layers, enabling flexible shape regularization across the input domain.

- **S3**. Beyond conventional supervised prediction tasks, the work extends to control/RL settings—demonstrating experiments on frequency control—which is a novel contribution.

**Weaknesses:**

- **W1.Rationale for probabilistic constraints is underdeveloped.**

While the attempt to impose constraints probabilistically is novel, the motivation is not sufficiently convincing. Framing monotonicity as a controllable “tendency” may be perceived as not providing a true guarantee. The paper does not adequately justify when monotonicity should be enforced probabilistically rather than strictly.

- **W2.Insufficient guidance on choosing $ \alpha $.**

Even if one believes monotonicity is a reasonable tendency, it is unclear what value of α should be used. Treating α merely as a tunable hyperparameter means the selection may react to data noise and thus may not yield a principled or reliable choice.

- **W3. Lack of certification when  $ \alpha = 0 $ .**

When $ \alpha = 0 $ , the method ostensibly enforces full monotonicity, but the paper does not provide a mechanism to verify that the learned model is globally monotonic. As noted in W1, claiming to guarantee monotonicity only “to some extent” is not fully persuasive. To strengthen the contribution, there should be a certification/verification procedure (akin to Certified MNN) that confirms global monotonicity when $ \alpha = 0 $ .

- **W4. "Easier to train" claim is ambiguous.**

The claim that the proposed method is easier to train than LMN, Constrained MNN, or SMNN is presented mostly as a post-hoc interpretation of outcomes. The paper should provide direct empirical evidence (e.g., convergence speed, failure rates, computational/gradient overheads) that demonstrates why and in what sense the approach is more trainable.

- **W5. Fairness of experimental comparisons is questionable.**

In the real-world experiment, $ \alpha $ is set to 0.1, which makes comparisons with fully guaranteed monotonic baselines potentially unfair. Not only methods that structurally guarantee monotonicity, but also benchmarks that—like the proposed approach—adopt a regularization-based strategy (e.g., Certified MNN, COMET) report performance only after conducting separate certification/verification steps to ensure monotonicity. Therefore, tables (e.g., Table 1 and 2) would be more meaningful if they reported results with  $ \alpha = 0 $  and global monotonicity guaranteed, so that performance is compared under the same guarantee level.

**Questions:**

- **Q1**. I’m curious about real scenarios where constraints hold only as a tendency. For example, in what kinds of problems might we expect a weak monotonic tendency of about 30%, versus a strong tendency of about 90%?

- **Q2**. Even assuming such tendencies exist, how should one determine the optimal value of $ \alpha $? Treating $ \alpha $ as a tunable hyperparameter may simply track data noise and may not yield a principled choice.

- **Q3**. In safety-critical domains (e.g., control) where monotonicity must be guaranteed, isn’t the proposed method arguably unsuitable in practice?

- **Q4**. If the method is configured with $ \alpha = 0 $, I’m not convinced the trained model will actually guarantee monotonicity. A verification/certification mechanism—analogous to Certified MNN—would strengthen the work. Have you considered adding such a procedure?

- **Q5**. A technical curiosity: how does the effective influence of $ \alpha = 0 $ change as network size or parameter count grows?

- **Q6**. When the dimensionality of the monotone variables increases, you sample uniformly over the domain and penalize violations. In practice, will the achieved violation rate on real datasets align with the target $ \alpha $ specified during training?

- **Q7**. I would like to see the trajectory of the dual variable $ \mu $ during training. Reporting this in the appendix would, in my view, improve the paper’s quality.

- **Q8**. For the Constrained Monotonic Neural Net and Scalable Monotonic Neural Net baselines, could you specify the exact network architectures used in your experiments? Beyond parameter counts, details such as per-layer widths (especially for layers handling different roles) would help readers interpret the results.

- **Q9**.  What are the differences between your update scheme and the approach in [1]? They seem related, and since [1] can also handle monotonic penalties, citing and comparing against it could strengthen the paper.

[1] Gupta, Akhil, et al., “PenDer: Incorporating shape constraints via penalized derivatives,” AAAI 2021.

---

### Official Review · Reviewer_mLnm · 2025-11-01

**Soundness:** 2
**Presentation:** 3
**Contribution:** 1
**Rating:** 2
**Confidence:** 4

**Summary:**

This paper formulates neural network training as an optimization problem, treating the training loss as the objective function and monotonicity as a constraint. The authors propose a method for training monotonic neural networks by considering monotonicity as a chance constraint, where alpha denotes the probability of violating the constraint. By adjusting it, the method provides flexibility to balance the trade-off between constraint satisfaction and prediction accuracy. In addition, by introducing an auxiliary variable t, the constraint becomes differentiable, and a primal-dual learning algorithm is proposed to solve the optimization problem.

**Strengths:**

- The introduction of the auxiliary variable t to ensure differentiability is well-motivated. This design allows gradient computation in a straightforward manner, making neural network training more tractable.
- The proposed method can be applied to traditional MLP architectures without requiring any structural modifications or architectural constraints.
- The paper provides an application example in power system frequency control, demonstrating the potential practical relevance of the monotonic neural networks.

**Weaknesses:**

I would like to point out only one weakness of this work; however, I consider it a highly critical one.Previous studies on monotonic function estimation treat monotonicity as a property that __must__ be imposed. In other words, the goal of estimating a monotonic function is to ensure monotonicity, while maintaining as much representational capability as possible, not to give users the choice to trade off between monotonicity and predictive performance. Therefore, the authors’ first claimed contribution, namely the __high flexibility__ of their approach, actually contradicts the fundamental motivation behind this line of research. If the authors could provide convincing real-world examples or empirical experiences demonstrating that such flexibility is indeed necessary or beneficial, I would be willing to raise my evaluation score.

**Questions:**

- Does having structural constraints in a neural network truly make training more difficult? Are there no cases where such constraints do not hinder learning?
- To enforce monotonicity over the entire input space, the model samples z values. If the number of samples N is not sufficiently large, wouldn’t it be difficult to achieve the intended effect?
- Since Uni(X) is also finite, can it really be claimed that monotonicity is enforced over the whole input space?
- Is there any specific reason why alpha was set to 0.1 in all experiments?
- Shouldn’t alpha be set to 0 (or at least a very small value) in order to guarantee monotonicity?
- Given that alpha is not zero, since the method employs a chance constraint, why are there still cases where the performance is worse than other methods?

---

### Meta-Review · Area_Chair_1JtP · 2025-12-31

**Summary:**

This paper concerns the problem of monotone classification. The Authors introduce a framework for training neural networks with a stochastic monotonicity constraint, where the level of violation is controlled by an additional parameter.

Unfortunately, the Reviewers pointed out several critical issues of the paper. The main ones are that the "partial" monotonicity is not well motivated and, even when the level of violation is set to 0, the method does not guarantee the model to be fully monotone. In addition, the empirical comparison with competing methods seems to be limited and unfair.

**Reviewer Concerns:**

Unfortunately, there is no rebuttal sent be the Authors.

**Reviewer Scores:**

The Reviewers would not change their scores.

---

### Decision · Program_Chairs · 2026-01-26

Reject